# Asian Medicinal Plants' Production and Utilization Potentials: A Review

**Sri Astutik [1,2,*], Jürgen Pretzsch [2] and Jude Ndzifon Kimengsi [2]**

[1] Cibodas Botanic Garden, Indonesian Institute of Sciences (LIPI), Jl. Kebun Raya Cibodas PO BOX 19 Sindanglaya, Cipanas, Cianjur 43253, Indonesia

[2] Institute of International Forestry and Forest Products, Faculty of Environmental Sciences, Technische Universität Dresden, 01735 Tharandt, Germany; tropen@forst.tu-dresden.de (J.P.); jude_ndzifon.kimengsi@tu-dresden.de (J.N.K.)

\* Correspondence: sri.astutik@tu-dresden.de

**Abstract:** Medicinal plants research in Asia continues to receive significant national and international attention, particularly concerning its multiple roles in poverty alleviation and health care support. However, scientific information on the institutional arrangements, the potentials of different medicinal plants production systems, and the utilization methods, remain highly fragmented. This incomprehensive information base shades the development of a comprehensive research agenda to improve the current body of knowledge, at least in the context of Asia. To address this impasse and propose future research perspectives, we systematically reviewed 247 journal articles, 15 institutional reports, and 28 book chapters. From the reviews, five key lessons are drawn: (i) Asian medicinal plant production systems demonstrate some dynamics, characterized by a gradual but continuous shift from wild gathering to cultivation, (ii) sub-regional variations exist with regards to the appreciation of medicinal plants potentials for traditional healing, modern healthcare and livelihoods support, (iii) knowledge on the effect of multi-scale institutional arrangements (formal and informal) on medicinal plant management practices is fragmented, (iv) very few studies dwell on the challenges of medicinal plants commercialization, particularly with regards to the role of middlemen, boom–bust cycle, raw material readiness, and product quality, and (v) law enforcement, benefit and knowledge sharing, and research and development should be prioritized to serve the interest of medicinal plants production actors. To further extend the body of knowledge on medicinal plants in Asia, we advance the need for empirical investigations on the performance of medicinal plants production systems and their contribution to livelihoods in diverse institutional contexts.

**Keywords:** medicinal plants; Asia; sustainability; utilization; potentials; institution; livelihoods; production system

## 1. Introduction

Medicinal plants (MP) significantly contribute to affordable healthcare and livelihood security, making them one of the most valuable non-timber forest products (NTFP) [1–3]. In this paper, the definition of MP follows Schippmann et al., Hall et al., and Slikkerveer who define it as those plants which are used for healthcare purposes in both allopathic and traditional medicine systems, and covers a wide range of species used including condiments, food, aromatic and cosmetics [4–6]. Traditional medicinal practices in Asia have existed from time immemorial; classical examples are *Ayurveda* (Himalaya, 4500–1600 BC), *Jamu* (Indonesia, 800 AD), *Traditional Chinese* Medicine/TCM (China, 3000 BC), *Sowa Rigpa* (Bhutan, 700 AD), *Kampo* (Japan, 500 AD), *Thai* medicine (Thailand, 1200 AD), and *Herbal Medicine* (Bangladesh, 4500–1600 BC and 900 AD) [7–13]. An inventory of medicinal

plants have been effectively carried out in several contexts [14–17]. Both the Bible and the Qur'an all underline the food pharmacy potential of over 240 species [18,19], all considered to be crucial in biodiversity conservation, livelihood support, trade, and the promotion of economic growth [20][21][22].

Traditional herbal medicines are rooted in indigenous knowledge systems. These cognitive systems play a crucial role in decision making with respect to the use of medicinal plants resources and are embedded in the lifestyle of the local community [23]. For instance, the use of rosemary (*Rosmarinus officinalis*) can be analyzed based on a botanico-historical, linguistic and statistical approaches, including historical documentary evidence [24]. Thus, MP that have been used traditionally for almost two centuries continue to feature in modern drugs [25–28].

At least 70% of the population of the developing world directly rely on traditional medicine for primary health care [29,30]. Equally, the industrialized nations indirectly rely on medicinal plants for their pharmaceutical products [31–33]. An estimated 25% of modern pharmacopeia and 18% of 150 top prescription drugs are plant-based [34,35]. China and India are two major international players from Asia in this regard [36]. Asian medicinal plants account for about 50% of export quantity and 45% of global earnings from traditional medicines [36]. They are utilized at the household level and for commerce.

As one of the significant bioresource centers of the world, Asia accounts for over 38,660 species of medicinal plants [37–42]; about 78 species are grown and commercialized, with China accounting for about 26 species [35]. Medicinal plant extraction and cultivation form an integral part of several Asian countries, including Bangladesh, China, India, Nepal, Pakistan, Myanmar, and Indonesia [43–49]. However, future research and policy interventions with regards to medicinal plants production and commercialization, and their contributions to the household and national economies of Asian countries remain unclear. The lack of clarity is rooted in the highly fragmented body-of-knowledge on medicinal plants production systems and utilization methods. This specifically concerns the institutional arrangements (formal and informal), and the potentials (production, utilization and commercialization) of different medicinal plants. To address this impasse and propose a future research agenda, this paper systematically reviews the current knowledge base on medicinal plants, with a focus on its institutional setting, production potentials and utilization, and commercialization.

The inspiration to prepare an article in this direction originates from the lead author's experience as a plant conservation researcher in the Cibodas Botanic Garden (Indonesia) between 2008 and 2016. During this period, she was involved in ethnographic fieldwork involving medicinal plants at the buffer zone of Meru Betiri National Park (1999–2000) for a period of six months, and with the indigenous people of Tau Taa Wana (Central Sulawesi) for a four-month period (between 2010 and 2011). The research was geared towards investigating the dual functions of medicinal plant use and its management for healthcare needs and livelihoods in community-based forestry settings [50,51] and the traditional healing potentials of local medicinal plant extraction [52]. These studies which support previous works on the fragmented nature of scientific literature on medicinal plants [53–57], contributed in shaping the interest to develop a review paper as a logical starting point. This review paper therefore sets a future research agenda on medicinal plants studies in Asia, with regards to empirical and analytical approaches to be employed on the subject. Specifically, the paper aims to review the status of medicinal plants in Asia with regards to its: (i) Institutional setting, (ii) commercialization, and (iii) production system potentials and utilization methods.

## 2. Materials and Methods

The review is anchored on the sustainable development concept and by adopting [58] and [59], the literature screening approach was conducted to define the inclusion criteria (Table 1).

**Table 1.** Explanation of the category used to determine the inclusion/exclusion of the selected literature.

| Category | Explanation |
|---|---|
| Asian medicinal plants | Research on medicinal plants carried out in Asia, especially South East Asia, South Asia, and China, or those involved in other plants' species observation, e.g., forest tree species. |
| Connection with the sustainable development (SD) concept | • Three pillars of SD are reflected in the literature, focusing on either the ecological and/or economy and/or socio-cultural aspects and/or institutional support. <br> • The purpose is to review the current management approaches and identify strategies to improve the management of medicinal plant resources. Emphasis on key words such as sustainable wild gathering, traditional knowledge, and CBD were relevant. |
| Time period | A majority of the selected papers were in English text and were mostly published after 1992—the post-Rio Summit period—considered to be a remarkable gathering that raised global consciousness on biodiversity management and Sustainable Development. Interest was also to map the evolution of scientific literature on medicinal plants, in the context of Asia. |

However, four relevant texts with at least an abstract that is written in English were retained. The search was conducted between November 2016 and December 2018 divided into three steps. First, the institutional arrangement to synthesize information on MP enabling environments. Second, the commercialization activity to describe the importance of MP for income generation and livelihoods, linked also to the socio-cultural context. Third, production and utilization potentials to acknowledge production systems and their benefits, by considering environmental circumstances. From these issues, keywords were derived and used for literature research.

Guided by the key search items, we compiled publications from the ISI Web of Science, Science Direct, Springer, Google Scholar, EBSCO, and Wiley (Figure 1), by using the keyword search "medicinal plants". Further search was employed using the keywords: "medicinal plants + Asia", "medicinal plants + local rules", "medicinal plants + local management", "medicinal plants + production systems", medicinal plants + traditional beliefs", "medicinal plants + state laws", "medicinal plants + regulations, "medicinal plants + international agreements", "medicinal plants + research and development", "medicinal plants + utilization methods", "medicinal plants + local actors", "medicinal plants + cultivation", "medicinal plants + conservation", "medicinal plants + sustainable use", "medicinal plants + sustainable wild harvesting/collection", "medicinal plants + commercialization", "medicinal plants + livelihoods", "medicinal plants + trade", "medicinal plants + income", "medicinal plants + value chain", "medicinal plants + economic importance", "medicinal plants + social", "medicinal plants + *sui generis*", "medicinal plants + traditional knowledge", "medicinal plants + gender", "medicinal plants + green economy", "medicinal plants + pharmacopoeia", "medicinal plants + CBD", "medicinal plants + CITES", "medicinal plants + Nagoya Protocol, "medicinal plants + Aichi Target". Most of the identified papers were confined to particular sub-regions, namely Southeast Asia, China, and South Asia and not central Asia and the Middle East. The review focustherefore, aligns with these regions. Based on this search approach, we finally arrived at 247 journal articles, 15 institutional reports, and 28 chapters of textbooks. Some retained papers were helpful in reporting several chosen analytical indicators (see Table 2).

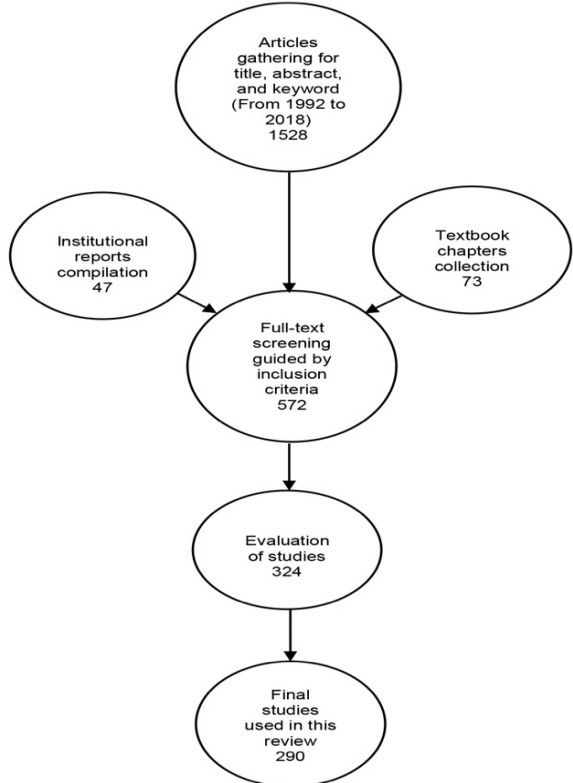

**Figure 1.** Schematic view of the literature selection process.

Subsequently, the literature was synthesized under three dimensions, namely, institutional setting, production potentials and utilization, and commercialization approaches. From the review, seven relevant indicators and their sub indicators were identified (Table 2) to be used in the systematic analysis [60–62].

**Table 2.** Review framework and indicators.

| Aspects | Indicators | Sub Indicators | Number of Literature |
|---|---|---|---|
| Institutional | Institutional framework, Socio-cultural dynamics | 1. Rules and structures | 16 |
| | | 2. Intellectual property rights and traditional knowledge practice | 23 |
| | | 3. Research and development | 16 |
| | | 4. Gender | 17 |
| Commercialization | Market requirement | 1. Product quality | 14 |
| | | 2. Distribution channel | 23 |
| | | 3. Financial benefit | 12 |
| Production potentials and utilization | Production system | 1. Wild gathering | 20 |
| | | 2. Cultivation | 33 |
| | Scale of usage | 1. Mixed-purpose | 56 |
| | Maintaining sustainable stock | 1. Conservation | 30 |
| | Impacts on ecosystem | 1. The local forest management system | 17 |
| | | 2. Ecological assessment | 20 |

# 3. Results

## *3.1. Institutional Framework*

The institutional framework discussed in this section encompasses rules and structures, intellectual property rights and traditional knowledge practice, and research and development.

### 3.1.1. Rules and Structures

Particularly, studies on access, use, and the management of medicinal plants are scanty in Asia. However, a few studies have shown that national regulations largely reflect global conventions, such as the Convention on Biological Diversity (CBD), the Nagoya Protocol (NP), Aichi Targets, and the Convention on International Trade in Endangered Species of Wild Fauna and Flora (CITES). Most of the literature revealed a disconnection between the existence of national regulations and their effective application. Informal rule crafting equally exists in these contexts. However, there is no clear information on the influence of both institutions in regulating access to and use of medicinal plants. This lack of clarity transcends the production system and equally includes commercialization and benefit sharing [53]. In Bangladesh and Nepal, for instance, uncoordinated cross-scale institutions and institutional bottlenecks hamper the functioning of medicinal plant systems [53,54,63]. Effective and sustainable regulations are required to guide medicinal plants production and commercialization systems around the Himalayas [64,65]. However, empirical studies need to reveal the conditions under which this should occur.

Nine countries, China, Korea, India, Indonesia, Malaysia, Myanmar, Sri Lanka, Thailand, and Vietnam, have documented their National Monographs for herbal drugs, while Pharmacopeia is found in Bangladesh, India, Indonesia, Sri Lanka, Thailand, and Vietnam [66,67]. In practice, TCM is more globalized and well-documented [67–70]. In general, there is increased interest by practitioners to implement good MP management practices—this suggests an urgent need for scientific investigations to inform such a process [71–75].

From the foregoing, it is clear that comprehensive information on the degree of rules enforcement (formal and informal) in guiding medicinal plants production, use, commercialization and management is needed.

### 3.1.2. Intellectual Property Rights and Traditional Knowledge Practice

The literature on Intellectual Property Rights (IPR) emphasize its sensitive nature in the medicinal plants production process; it is pertinent for information, knowledge, and commerce [76]. Six papers indicate that medicinal plants have contributed significantly to modern drug discovery [26,31,77–80]. Pharmaceutical research and industries now prioritize research on (1) new chemical component discovery, (2) know-how of production, and (3) product trademarks [81].

Basically, intellectual property (IP) should firmly consider that local people have managed and conserved medicinal plant resources for a long time [82–84]. The dilemma of linking IPR and traditional medicine remains unaddressed [81,85,86]. Besides, traditional knowledge protection often involves ethical and cultural sensitivities. So intellectual property rights (IPR) systems are unsuitable to apply in this regard [87,88].

In terms of traditional knowledge access, Cordell mentions that it might face complexities and bureaucracy with variations from country to country [89]. In the context of Southeast Asia, Antons and Asma and Talaat discuss the challenges of implementing a community-based model for traditional knowledge and genetic resources due to conflict of interest [90,91]

The literature on medicinal plant cultivation and commercialization related to IPR remains scanty [61,92,93]. Ethnobotanical knowledge also affects medicinal plants' consumption patterns, especially for urban and peri-urban people [5]. This affects the inter-generational transmission of knowledge. Only a study by Singhal suggests four methods of transmitting medicinal plants' knowledge to upcoming generations, namely, learning by observing, learning by doing, learning by sharing, and transfer of IPR [94].

### 3.1.3. Research and Development (R&D)

Some articles underline that research, development, and financial investments are urgently required for the scientific and knowledge enhancement of MP. Many R&D institutions focus on core competence, in line with their bioresource advantages [95]. Generally, all of them apply a multi-disciplinary approach to determine potency and to optimize existing resources, by improving regional/global networking, human resources, infrastructure, expertise exchange, capacity building, and government policies [96–98]. The two recurrent aspects under R&D for medicinal plants in Asia are described as follows. Research

focusing on the sustainability of production systems is still lacking, even though major supply relies on wild harvesting [99]. A few studies on wild gathering focused on how to improve the yield, rather than to determine its impacts on the population level. A few papers reveal good examples, i.e., (1) using 70% hand plucking technique in harvesting *Embelia tsjeriam-cottam*, (2) collecting 25% of all plant phases of *Rheum acuminatum* and *Rheum australe* in a rotation of 3–5 years, and (3) non-destructive methods applied on *Andrographis paniculata*, *Phyllanthus emblica*, *Terminalia arjuna*, and *Terminalia bellerica* [100–103].

Mostly papers of ethnobotany and ethnopharmacology are still prevalent in Asia, emphasizing local herbal knowledge and species uniqueness, such as Pakistan, Bangladesh, Myanmar, India, Philippines, Indonesia, Bhutan, Nepal, Lao PDR, Sri Lanka, China, Vietnam, and Malaysia. These also direct to many research on potential herbal-based modern medicines [28,96,104,105]. However, India and China are the only established model of integrating traditional and modern medicine, among other countries [106]. A Bhutanese traditional medicine species, *Meconopsis simplicifolia*, is observed for antiplasmodial activity [107], while in Pakistan, at least 15 plant-based compounds are registered for drug development [108].

The trend of Asian medicinal plants research and their geographic distribution (Figure 2) indicate that South Asia accounts for the highest number of studies on this subject, followed by South East Asia and China. The trend is relatively similar for the three fundamental research carried out in this region. Meanwhile, most of the studies focus on production potentials and utilization, followed by the institutional setting and socio-cultural dynamics, and commercialization.

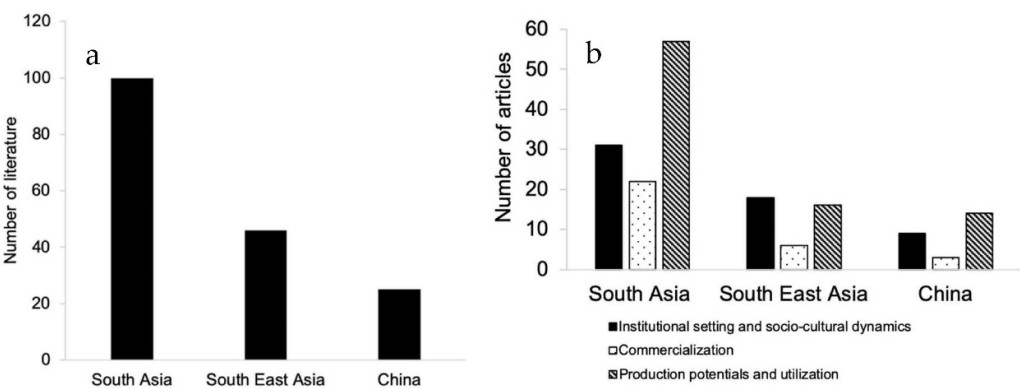

**Figure 2.** The trend of research issues on Asian medicinal plants based on (**a**) geographic range, and (**b**) research focus.

### 3.1. 4. Socio-Cultural Dynamics

Medicinal plants extraction or cultivation is labor intensive involving many actors along its production and market chains—women, men, and sometimes the children [109,110]. Gender roles are not fixed. Sometimes, men are involved in harvesting, carrying, drying, trade, and transport, while women work with sorting and packing [111]. Males frequently collect plants in remote areas, while women collect from accessible and nearby areas [55,112]. For example, in Swat District Pakistan, collection activity involves around 75% of children for a seasonal job, 21% women, and 4%

older people [110]. Meanwhile, in trading activity, the collectors are dominant (93%), while the rest are dealers (7%). In Pakistan and the Philippines, women play a significant role in the treatment of ailments at the household level, while the men are responsible for MP collection [113,114].

Regarding knowledge capacityin Southeast Asia, women are well-known users of 2000 different species in almost 5000 combinations for healthcare purposes [115]. In South and Southeast Asia, a total of 1875 species are applied for menstrual and reproductive health issues [116]. For example, in Lao PDR and Thailand, between 55 and 79 species are utilized in women's healthcare, respectively [117,118]. In Malaysia, postnatal care treatment makes use of at least 128 species prepared in numerous forms [119]. In Indonesia, almost 10 to 30 herbs can be found in one type of Jamu Madura, as a reproductive remedy for women　[120]. These are in accordance with women's task, which is to frequently receive the responsibility to serve as health providers as part of their domestic role [121].

Women's role as the decision makers has grown in the management of medicinal plants. For example, research by Olsen and Bhattarai underline that women also participate in marketing at the household level, but most traders and wholesalers are the men [109]. According to Torri, in Indonesia, commercial activities of Jamu and herbal-based cosmetics mostly conducted by women producers can increase the revenue up to 20% and transfer health-beauty knowledge among actors [122–124] Also, some other benefits encompassed less dependency to external health services, better health access and market for women, and more health agencies.

Gender roles in the production, utilization, and commercialization of medicinal plants remain nuanced; it is not clear where either men or women are the predominant group engage in this activity. This suggests further empirical studies to clarify gender roles in the medicinal plant process in Asia.

The institutional framework identifies the following aspects requiring urgent research attention: (1) An investigation of the conditions under which rules and regulations (formal and informal) can be successfully enforced at multiple levels, to regulate medicinal plants production, use and commercialization. (2) The dilemma of linking IPR and traditional medicine needs to be unbundled. (3) The inter-generational transmission of medicinal plant knowledge processes needs to be studied. In a nutshell, the institutional framework, from domestic to global levels can be seen in Figure 3.

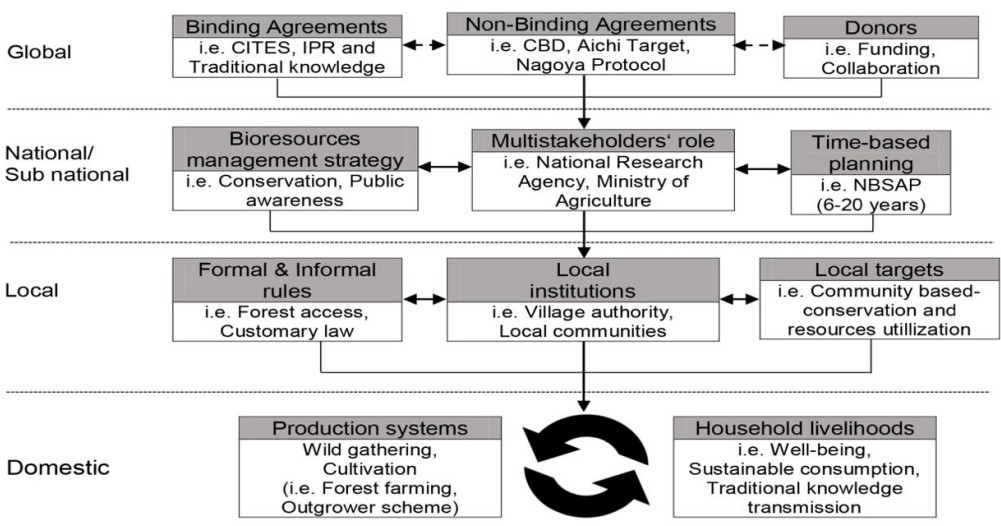

**Figure 3.** Institutional framework of Asian medicinal plants management and conservation. NBSAP: National Biodiversity Strategy and Action Plan.

*3.2. Commercialization*

This section discusses three aspects of market requirement, namely, product quality, distribution channel, and financial benefits.

3.2.1. Market Requirement

The continuity of raw material supply due to a growing market is a crucial point in medicinal plants marketing. This specifically concerns the availability of raw material and agronomic practices, which are inseparable with post-harvesting methods, proper market channel, and supply chain [93,125]. Furthermore, the phenomena of boom–bust cycles are considered as the most common challenge for NTFP, like medicinal plants [34,126]. It indicates an unstable condition, where the beginning harvest is followed by decreasing resource availability. It increases transaction costs, creates hostile market situations, and a specific product dependence due to unpredictable change in demand [127,128]. There are some beneficial strategies to encounter those challenges mentioned above, such as cultivation, value chain upgrading, quality assurance, and networking enhancement [34,99,127,129]. In terms of cultivation, many papers show that growers can work on either private or state lands depending on the case types, either for pure commercialization or community empowerment projects (see Appendix A).

- Product Quality

Product quality is critical for the safety and efficacy of botanical medicines, and also for consumer confidence, credibility, and risk reduction [68,130,131]. Fundamentally, quality improvement and market access are challenging in the medicinal plants' sector, from wild harvesting or cultivation to the end users [132,133]. Three sets of quality standards, namely International Standard for Sustainable Wild Collection of Medicinal and Aromatic Plants (ISSC-MAP), Good Agricultural Practices (GAP), and Good Manufacturing Practices (GMP), are firmly requested in the global market [134–136]. So far, in Asia, China and Japan have established Good Agricultural and Collection Practices (GACP) for medicinal plants [134]. A success story of wild harvesting certification on *Schisandra sphenanthera* in China can be a good example in applying this standard [137]. How these international standards are framed in national guidelines and how they are further linked to local practices remain unclarified.

Product quality is linked to postharvest handling. For instance, in the case of Nepal, 20 years ago, the export of 36 unprocessed products to India was done without storage, transport, and grading [138]. Recently, mostly three types of products can be found, namely essential oils and extracts, semi-processed products, and finished goods [139]. In Nepal, by incorporating postharvest treatment and value chain upgrading, the bay leaf farmers organized within the cooperative system can gain a trifold increase in price [140]. In India, Choudhary reveal that by upgrading the value chain of Indian bay leaves, local people could register an increase in income, leading to improved management of forest reserves [141]. On the other hand, Banks stressed that the perspective of actors about postharvest system depends on their own interests, such as high yield for growers, high prices for marketers, or safe products for customers [142].

- Distribution Channel

There is a limited number of value chain studies, including markets on medicinal plants [143,144]. Based on the available research, it can be seen that two groups of actors vary depending on the products and market chains, they are upstream and downstream members. Upstream members encompass input suppliers, primary producers, processors, brokers, and traders, while downstream level comprises manufacturers, distributors, herbal doctors, retailers, and consumers [47]. In Pakistan, the structure is more simple, involving collectors, middlemen, traders, and exporters [110]. In China, the simplest chain among ten models is carried out by e-commerce based and integrated firms, while the others comprise 2–4 actors such as farmers, middlemen, processing firms, wholesalers, and retailers [145].

In general, many authors emphasize that middlemen tend to dominate the chain. This dominance leads to the inflation of prices and margin inequality [146,147]. However, market change-drivers also need to be considered, such as changes in demand, infrastructured development, and government interventions [148]. For instance, in Bangladesh, the profit margin varies among actors; they are middlemen (59%–139%), wholesalers (22%–90%), and processors (109%–358%) [47]. In Nepal, net margins for traders are lower (−5%–20%) than those for central wholesalers (25%–36%) [143]. In India, middlemen earn at least 26 times more than cultivators, and their profit margin is higher (20%) than that of wholesalers (10%) [149]. In Pakistan, the price in the international market is 2–12 times higher than that of the collectors or farmers [147]. In Indonesia, the price of the Indian Screw tree (*Helicteres isora*) is five times higher than in herbal industries [150].

According to Belcher and Schreckenberg, and Jensen, the success determinants of medicinal plants commercialization as a part of NTFP consist of some factors, such as: (1) the product concerned and its characteristics, (2) the markets, (3) demand factor, (4) risks and uncertainties, (5) integrated value chain and mechanisms to counter overharvesting, (6) national policy, (7) livelihood strategies considerations, and (8) quality and quantity improvement [126,151]. Therefore, a better understanding of value chain and market analysis could help to establish future intervention scenarios [152].

- Financial Benefits

Some studies associate income change with medicinal plants commercialization. For instance, in Nepal, local people obtain between USD 123 and USD 121, as brokers of their household; this contributes to 41% of their income and 3%–44% (average of 12%) in the alpine mountain of Himalaya [56,61]. In India, the total value of two cultivated species, *Saussurea lappa* and *Picrorhiza kurrooa,* account for USD 11,000/year [153]. In Central Himalaya, almost 10% of rural households are engaged in commercial wild crafting [154]. In China, local people obtain an average income of around 1.4 to 9.1% [154,155]. In Indonesia, the total profit of cultivated *Curcuma xanthorrhiza* range from USD 53–158/ton [156], while in Pakistan, the total revenue collected is USD 353,045 for *Morchella esculenta* and USD 353,045 million for 23 other important species [147]. Producers in Bangladesh witness an increase in profit ranging from 30%–130%. In Vietnam, this contributes up to 11% of household income [47,157].

Research on price is still limited; and most studies show that price is diverse depending on markets and product quality. In Nepal, harvesters achieve net margins between 34%–55% of the Indian wholesaler price [143]. In China, the cost of goji (*Lycium barbarum* and *L. chinense*) in high-quality markets (USD 11.0–20.5/kg) is higher than in conventional markets (USD 6.3–9.5 /kg) [145]. A study by Booker et al. point that value-added treatment can be more beneficial due to different processing methods [144]. For instance, the price of turmeric increases almost tenfold from food grade powder (USD 19.3/kg) to encapsulating product (USD 277.8/kg). Besides, the price of 20 species of TCM in the UK supply level is 4–40 times higher than for China.

Market requirements, regarded as the quantity that is sufficient to ensure stable market supply, needs to be well established. At the moment, literature still demonstrate significant deficits. Besides, value chain analysis including governance, actors and upgrading options in different medicinal plants production systems, need to be fully investigated in the Asian contexts. Figure 4 illustrates the general market structure involving downstream and upstream actors.

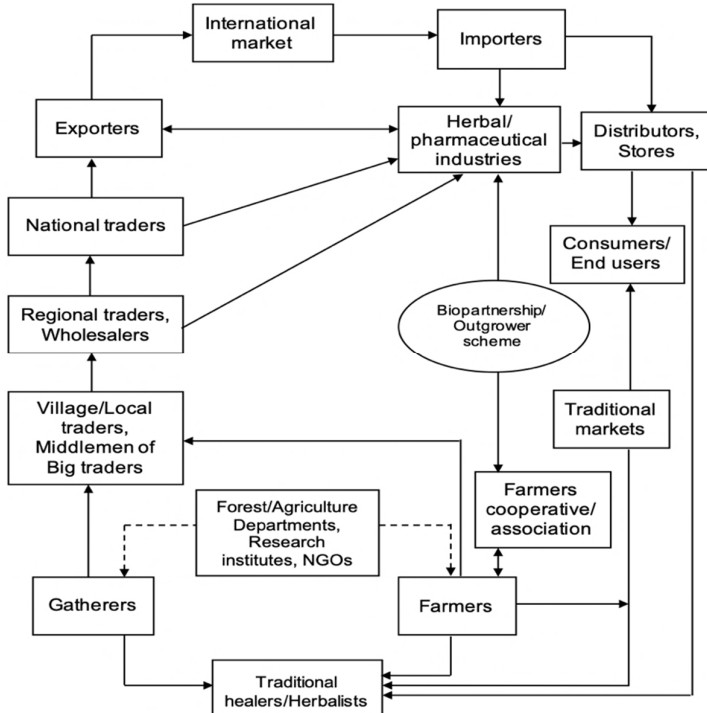

**Figure 4.** The general market structure of Asian medicinal plants.

*3.3. Production Potentials and Utilization*

The production potentials and the utilization of medicinal plants are reviewed under three perspectives, namely, the maintenance of sustainable stock, production systems, and the scale of usage.

3.3.1. Production System

- Wild Gathering

Most of the papers indicate that the key actors involved in wild gathering are closely related to medicinal plant use either for subsistence or for commercial purposes . A vivid documentation of the collection of raw materials from the forests by traditional healers to treat ailments in Bangladesh, Myanmar, and India is provided [11,158,159]. A few studies indicate the income benefits derived from medicinal plants' collection [65,160]. For example, in Bangladesh, the purpose was for own consumption (63%) and commerce (37%) [161]. Conversely, in far-west Nepal, almost 67% of the harvest was sold and the rest is for domestic use [162]. In Malaysia, at the household level, communities living in the forest of Jagoi area obtain around 715.38 USD per year of medicinal plant use [163].

Studies have further documented overexploitation and unsustainable harvesting as a key issue in the wild gathering system (Appendix A). In Indonesia, at least nine species in Java are endangered due to extensive extraction, while 21 species in Kalimantan are threatened [164,165]. A rise in the trade of several threatened species has been observed for India and Vietnam [157,166]. Overharvesting of this natural system is linked to unclear access and use rules in most of these traditional systems. Angelsen et al. emphasize that exclusionary conservation policies could threaten the livelihood of local forest people [167]. Homma proposes a model of extractive resources as an economic cycle determined by multi-factors such as wild stocks availability, development and environmental policies, socio-economic attributes, scientific and technological development, migration tendency, and labor markets [168]. The shortcomings linked to cultivation need to be recognized such as declining stocks, low prices, low profit, income elasticities, a synthetic substitute

discovery stimulation, a disproportionate changing demand, and its less competitive nature compared to other sectors of the economy. In sum, literature contends that access rules for wild gathering, the performance of this production system, vis-à-vis other systems, remain largely unclear. This suggests a strong need for a comparative analysis of the economic performance of medicinal plant production systems.

- Cultivation

Many studies show that currently, MP cultivation has grown in some countries of Asia to address the conservation of valuable species, generate income for local people, and support regional economic development (Figure 5 and Appendix A). In this case, most of the literature on cultivation originate from South Asia, while the transition from wild gathering to cultivation is linked more to South East Asia and China. However, only about 3.3% of most medicinal plants are cultivated, while the remaining proportion is derived from wild gathering.

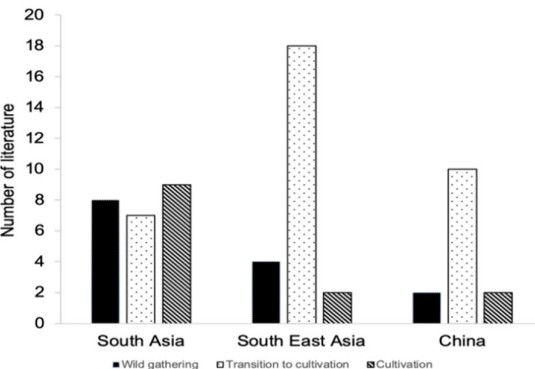

**Figure 5.** The trend of transition to cultivation in Asia.

Information on the trend of MP practice is relevant in guiding future supply and material needs alongside the maintenance of its ecological and economic values [169–171]. Recent statistics show that only a few countries such as China, India, Indonesia, Nepal, Thailand, and Vietnam, produce and commercialize medicinal and aromatic plants [32]. Cultivation can optimize yields and ensure high product quality [172]. Besides, domestication and cultivation also reduce wild gathering pressure [173,174]. In terms of commercialization, the main concerns consist of the availability of agro-technology and the search for remunerative markets [175]. Thus, cultivation types can be considered based on an economic importance scale (from low to high), , the possibility for immediate cultivation, or conspicuous products [168]. Additionally, some advantages linked to improve wild harvesting exists, including low cost of production, increased return to land and labor, product quality, increase in supply capacity, and manageable price and production factors.

With respect to maintaining supply and promoting the growth of herbal industries, the bio-partnership approach is adopted to explain an equal partnership between the local community (small growers, women, tribal people) and herbal industries [53]. There are not so many studies about bio-partnerships; most cases are from India. For instance, the Ayurvedic company purchased at 10% above the market price of three valuable species, *Aconitum heterophyllum*, *Aconitum atrox*, and *Saussurea costus* [111]. A growers association promoted some scarce species such as *Catharanthus roseus*, *Piper longum*, and *Rauvolfia serpentina* to the national herbal industries and negotiated a fairer price [176].

Scientific evidence also holds that combining cultivation and conservation can provide significant impacts on economic gain and local livelihood improvement. For example, the community-based approach has significantly improved planting materials' production, local livelihood, sustainable harvesting and marketing, skills, and knowledge in India and Nepal [46,49,177]. In China, cultivated species contribute 10%–70% of the household income for local people

[178]. In Nepal, private farmlands contribute about 32% of the raw material produced and 20% of cash income [179]. Cultivation is usually seen as an approach to improve local livelihoods [177,180–182].

Research focus in this regard is seen from the following perspectives: First, the integration of conservation techniques and cultivation practices to enhance preservation and production [58,134,183,184]. Second, that the practice of organic farming is a panacea for economic and environmental challenges [58,185]. Third, the integration of medicinal plants into the agroforestry model in a few Asian countries needs research attention related to cultivation and marketing issues [186,187]. Fourth, home gardens are useful for domestic use and for sale [10,44,157,188].

In spite of a beneficial return, cultivation also brings some drawbacks as mentioned by some studies. They consist of ecological and socio-economic aspects. For instance, [99] explain that it is challenging to cultivate some species due to ecological reasons. On the contrary, Williams and Ones propose a bioeconomic model on the cultivation of overexploited species, *Chamaedorea ernestiaugusti,* to show its significant effect on increasing wild harvesting pressure [180].

### 3.3.2. Scale of Usage

Based on the existing literature, the scale of usage is important to distinguish between subsistence and commercialized medicinal plants in Asia. So far, it is considered as a mixed-purpose of raw material obtained through wild gathering or cultivation (see Appendix A).

In Asia, generally, medicinal plants have been cultivated for herbal medication, food, and cosmetics. Recently, as marketable goods, medicinal plants demand is witnessing a steady increase. Close to 100 articles maintain that herbal medicines are considered as one of the bio-based dominant sectors and the demand is increasing significantly. For instance, India and China use at least 8000 products [83,189,190]. Herbal industries contribute to national economic growth per year as follows: India (USD 327.7 million/0.014%), China (USD 18 billion/0.16%), Malaysia (the USD 800 million/0.27%), and Indonesia (USD 297 million/0.032%) [189,191–193].

In the South Asia region, most people rely on medicinal plants resource for subsistence and traditional medication. Both are vigorously practiced in routines such as collecting medicinal plants for *Ayurveda* and *Unani*; some proportions are traded [194]. Meanwhile, in South East Asia, people tend to use medicinal plants for curing ailments based on a robust traditional knowledge and practice; the reverse is the case for China. Nevertheless, in South Asia and South-East Asia, medicinal plants are commercialized by small and large-scale industries.

### 3.3.3. Maintaining Sustainable Stock (Conservation)

Generally, medicinal plant utilization without conservation efforts lead to species scarcity, especially for endangered species [53,195–197]. Several articles highlight the importance of attitude change in promoting conservation [198–201]. The integration of traditional and local ecological knowledge in forest management, by looking at the diversity of knowledge, practice, and belief systems is required [202]. The case of Joint Forest Management in India which enhances collaboration between communities and forest administrators should be a priority alongside rising local awareness and knowledge of rights [203]. Several articles on community forestry in Nepal show that local people utilize at least 100 species of MP, with 34 of such species having potentials for cultivation and subsequent commercialization [61,204–206]. Additionally, an apprenticeship method under the guidance of *Amchi* can conserve traditional knowledge in the future [207,208]. On the other hand, China's Belt and Road Initiative should be seriously taken into account with regards to the potential negative impacts on biodiversity management including Asian MP. The highway can potentially grant more access into protected areas, contribute to habitat devastation, and lead to the growth of invasive alien species. Although a core value of biodiversity concept sounds promising to be applied, an integrative discussion within global and regional stakeholders is required [209,210].

With respect to connecting conservation practice and people, global guidance on information and research, policy and legislation, conservation strategies, sustainable production, equity, awareness, training, and capacity building are emphasized [211]. Opposing ideas regarding the application of either a holistic approach [212] or the development of detail parameters, namely native

and conservation status, economic and ethnobotanical values, global and national distribution, legislation and threat assessment have been raised [213].

The in situ method considers three aspects strongly, namely, ecosystem protection, viable population maintenance, and natural habitat management [214]. In this case, many studies tend to emphasize community-based approaches due to a potential conflict arising between the government and local people [46,215].　This also can lead to a positive attitude for the local people [216]. For instance, in China, the protection of the sacred sites of Tibetan mountain containing 206 species (113 endemic ones) contributes to preserve native biodiversity and local knowledge [217]. Ex situ is usually established in the form of living collections and high genetic variation outside the natural habitat, such as botanic gardens, field gene banks, seed banks, and in vitro storage [218–220]. The challenges could be linked to costs, risks, and other scientific aspects depending on the species, the method, and the time range [221].

Most of the literature indicate practical steps towards the conservation or maintenance of medicinal plant stocks. However, these studies do not point to the performance of different production systems and the maintenance approaches to employ, including the link between sustainable medicinal plant management and livelihoods.

3.3.4. Impacts on the Ecosystem

- The Local Forest Management System

In almost all countries considered in this review paper, gathering medicinal plants as NTFP is done through collector groups. For example, at least 95% of medicinal plants used and traded are harvested from the wild stock, and professional collectors who do so for marketing purposes are responsible for most of the damage [194,198,222]. On the other hand, some papers indicated that wild gathering and cultivation positively affect society, and medicinal plants in the context of forest utilization. Some customary rules can be effective local mechanisms to control forest from the over-exploitation of NTFP, due to social harmony circumstances [204,223–225]. It is in line with Gautam and Watanabe (2002) who noted that on the global scale, local people have applied ecological knowledge in forest management including silviculture of non-timber forest products [226]. In addition, efforts in sustainable harvesting and cultivation can bring positive effects to community members and medicinal plants' availability as well [227–230]. For instance, in Northwest Pakistan, re-growth forest and reforestation can improve medicinal plants' abundance, whereas the density of ten species is significantly correlated to increased prices [231].

With respect to wild gathering, community forestry schemes seem to be a popular approach; this is seen as a good model involving local initiatives in South Asian countries such as Nepal, India, and Bhutan [61,232,233]. For example, Van Panchayats in India facilitated by state and non-state actors, issues harvesting permits in the reserve forest. In Nepal, forest user groups have shown a reference model of managing and harvesting NTFP, including medicinal plants [61]. In the context of forest access, some studies noted　that local wisdom can help to control destructive impacts and to promote stakeholders' participation as well.

- Ecological Assessment

Many studies show the importance of ecological assessment in the sustainable management of MP [234–236]. In this paper, we found eight examples, such as: (1) Rapid inventory of medicinal plant population in Sri Lanka, (2) rapid vulnerable assessment of Tibetan medicinal plants, (3) assessment of a threatened high-value medicinal plant, *Swertia chirayita* for conservation purpose in India (4) using IUCN guidelines to determine threats to selected endemic species in Kashmir Himalaya, (5) distribution assessment of threatened species on hotspots in China, (6) questionnaire survey for screening endangered species in Bangladesh, and (7) studying CITES-listed species to reveal the international trade impacts, and (8) ten years monitoring of threatened medicinal plants in Himalaya [41,227,237–242].

In general, research and actions on invasive alien species (IAS) still need to be enhanced in Asia [243–245]. Despite undeniable environmental impacts of IAS [246], a few papers indicate its beneficial role for MP sources. For instance, in Bangladesh, at least 17, 39, and 43 species are used for curing many ailments in several protected forests areas [247–249]. In China, the second numerous usage of IAS is for MP sources [250]. Approximately 111 species are used for health care purpose in India [251].

Mostly, ecological assessment approaches are taken into account to investigate anthropogenic causes and propose applicable future management. These options also need to consider the drawbacks and benefits. The discussion on ecosystem impacts presents several aspects which still still require scientific investigation. For instance, an ecological assessment does not reveal the performance of different production systems. This has to be investigated to enhance knowledge on medicinal plants in Asia. Secondly, the level of organization of collector groups is not fully determined. Thirdly, the role of these groups in managing the different production systems remain unclear.

In a nutshell, the review on MP production potentials and utilization suggest that several aspects remain unclear and deserve further research attention. These include: (1) Access rules for wild gathering and the management of different medicinal plants production systems. (2) A comparative analysis of the performance of different medicinal plant production systems, in a bid to develop guidelines for the promotion of profitable and conservation-friendly systems.

## 4. Discussion

In accordance with the review of literature on Asian medicinal plants, three broad research issues beg for further clarification.

*First; Institutional perspectives*—Three interlinked aspects have not been addressed. Firstly, the conditions under which rules and regulations (formal and informal) can be successfully enforced at multiple (local, sub-national, and higher-national) levels, to regulate the production, use and commercialization of medicinal plants are needed. Studies that link medicinal plants and related international agreements are scanty. This information is needed to reveal the role of stakeholders in tackling the depletion of wild herbal resources, enhance power and knowledge-sharing between state and local actors and institutions, and potential/real benefit-sharing with multiple actors. With regards to global public health standards, there is a need to explore its application in the context of national governments [66,252,253].

Moreover, in-depth studies on institutional aspects are required, focusing on MP governance and management systems, actors, resource systems and resource units, interactions and outcomes. Ostrom argues that natural resource utilization requires a good understanding of the interconnection among related factors, such as socio-economic, political, and ecosystems frameworks [254,255]. Therefore, we suggest that future studies on medicinal plants should discuss how actors' roles can be established at the level of user groups, how far/close management innovation can sustain product quantity and quality, effectiveness of integrating ecological assessment and access/benefit sharing policy, and how external forces can affect socio-economic institutions, community empowerment, and market opportunities.

Subsequently, the unbundling of the dilemma to link intellectual property rights and traditional medicine is required. Regarding IPR, traditional knowledge protection can be aligned with transmission efforts to the youths engaged in different production systems. The recognition of local knowledge and practices through medicinal plants could serve as a springboard for medicinal plant-based livelihood, fair bioprospecting, and modern therapeutics as well [67,179,256]. Some global agreements on IPRs do not align with the perspective of indigenous and local people especially regarding "a bundle of spiritual and nature relationship versus a bundle of economic rights" [257]. Therefore, a study on the perception of indigenous/local people might reveal why IPR is needed, and the extent to which traditional medicinal knowledge can be commercialized.

Lastly, the inter-generational transmission of medicinal plant knowledge processes and their implications for production systems and economic benefits thereof are required. However, knowledge on the sustainability of MP production systems is lacking. Therefore, we recommend that

research on medicinal plants could be incorporated in the rural development context, especially in South East Asia and China. This will emphasize on sustainable livelihoods, value chain analysis, market development, community empowerment, institutional capacity enhancement, and policy considerations. Specifically, sustainable harvesting issues could be connected to poverty reduction and well-being due to their significant potentials for income generation and production capability. The actual and potential contribution of MP production systems needs to be fully uncovered. Further, synergistic research on advanced technology and production systems could open up opportunities to explore the specific locality of raw materials associated with local knowledge.

In the case of gender, a nuanced perspective exists with regards to gender roles in the production, utilization, and commercialization of medicinal plants. Studies on connecting labor, gender, and production systems are required. Such studies should seek to answer questions related to the extent to which time and labor allocation can affect production systems performance, and how it could lead to changes in the household economy. Few studies focus on linking gender roles in household decision making with regards to medicinal plants harvesting and use [258]. Also, very few studies indicate that women lead self-help groups, cooperatives, and other local economic institutions [259,260].

*Second; Commercialization aspect*—Regarding market requirements, literature still demonstrates significant deficits. Research related to contrasting species and the volume in supply and demand at the global level is scanty . For that reason, closing this gap will help Asian countries to determine their production strategies and species priority. Moreover, studies focusing on boom–bust cycles as an issue   in medicinal plants' production and marketing are also lacking. MP' commercialization follows the production to consumption system, involving many actors and markets [126,261]. Studies should indicate how raw material stock alongside postharvest handling can lead to price fluctuation, how the community group approach can raise its bargaining position, and what pattern of partnership can shorten the chain to assure a market for raw materials.

Finally, value chain upgrading options in different medicinal plants production systems need to be fully investigated in Asian contexts. It can be inferred that specific studies on MP value chains are rarely found. These studies can portray the importance of actors' role in the chain, connections across actors, and job opportunities for deprived groups [47,109,149]. This will promote a better understanding of the enabling environment for end markets and priority actions of intervention. Thus, medicinal plants commercialization should be given priority by multi-stakeholders, as a primary determinant of its utilization for income and job provision, profit sharing, product varieties, technology and knowledge, skill improvement, and collaboration [57,149,262,263]. Following Poschen et al., future value chain studies could examine research, mapping and analysis, or development of MP [152]. Further studies concerning the relationship between production systems and household livelihoods also need to be expanded. These   might contribute to develop interventions on policy, program, management practices, and organization for livelihoods enhancement.

*Third; Socio-ecological context*— The performance of different Asian medicinal plants production systems and their links to livelihoods needs to be investigated. With respect to the studies mentioned above, wild gathering and cultivation are complementing each other in terms of market demand; both possess some limitations. Cultivation can counteract inescapable wild gathering activity, but wild gathering does not support cultivation. It is in line with Schippmann et al. who explained based on the perspective of species and ecosystem, market demands, and people [99]. Explaining the benefit of combining both production systems needs to be explored with regard to either local or commercial use or mixed-purposes. Thus, a comparative analysis of some varieties of production systems is required to further understand the management practices and outcomes alongside their contribution to livelihoods [186].

The level of organization of collector groups at different levels from production to marketing is not adequately mapped. Future studies on MP actors should link the production to consumption system. This will reveal who should and what can be done along the chain from producers to consumers in the domestic/global market [126]. These complex processes might be connected to the location, the

nature of the products, the processing stage, and the customer requirements. In the meantime, local initiatives, regional, and national policies are inseparable to scrutinize wild stock supply versus market demand, considering a conflict of use, cultural importance, livelihood portfolio, partnerships, and stakeholder consultations. A better understanding of the connection between natural resources and people is necessary [21,264]. The role of actor groups in managing the different production systems remain unclear.

Thus far, in this paper, we establish that the cultivation trend is increasing in most countries of Asia. This indicates also a shift in the production systems from pure extraction to manageable resources. The motives might be linked to some factors such as conservation, local economy, reforestation, and industrial needs (see Appendix A). Currently, cultivation is an essential strategy for conserving and maintaining sustainable natural stocks, but, in fact, just a few medicinal plants are cultivated [32,195]. Massive cultivation can control the natural resources by managing wild harvesting and subsequently, it can assist medicinal plants survival in its natural habitat [21,265–267]. Rigorous research on the raw material's journey to finished products involving socio-economic and ecological aspects should be examined.

In general, the challenges faced by Asian medicinal plants management lie on the policies synchronization and its implementation in all scope, at the global, national, and local levels (Figure 3). On the other hand, the survivability of MP systems at the local level needs urgentattention . Local solutions become the prompt options, which are sometimes not parallel with the rules at higher level. So, dialogue and multi-stakeholders' involvement are required to harmonize these interests. In terms of market structure (Figure 4), gatherers and farmers often face a similar problem of market access. Traders and middlemen frequently play dominant role in the chain. Hence, cooperative and partnership with herbal industries can be an alternative. Lastly, environment and resources availability concern are indicated linked to over harvesting, habitat depletion, and local knowledge erosion. Even though the trend of cultivation is growing, the supply and demand gap remains problematic. Overall, the sustainable management of Asian medicinal plants is inevitably related to three essential pillars, as mentioned above (see Table 2 and Figure 6). Therefore, in this paper, we propose that each interconnected pillar specifically generates critical issues to be considered for future empirical studies. First, connecting production and utilization to the commercial system can continue to well-being improvement. Therefore, focus should be on the following aspects: sustainable livelihoods, value chain development, product upgrading, market access, and partnership. Second, MP commercialization tied to formal and informal rules and socio-cultural factors leads to fairness and equity. Hence, three basic principles are necessary to be employed namely, benefit-sharing, community empowerment, and international agreements. Third, connecting MP production and utilization to the institutional framework requires long term and simultaneous assurance in providing goods and services. Consequently, four elements are crucial; they are gender roles, local knowledge transmission, sustainable wild harvesting regulation, and GAP.

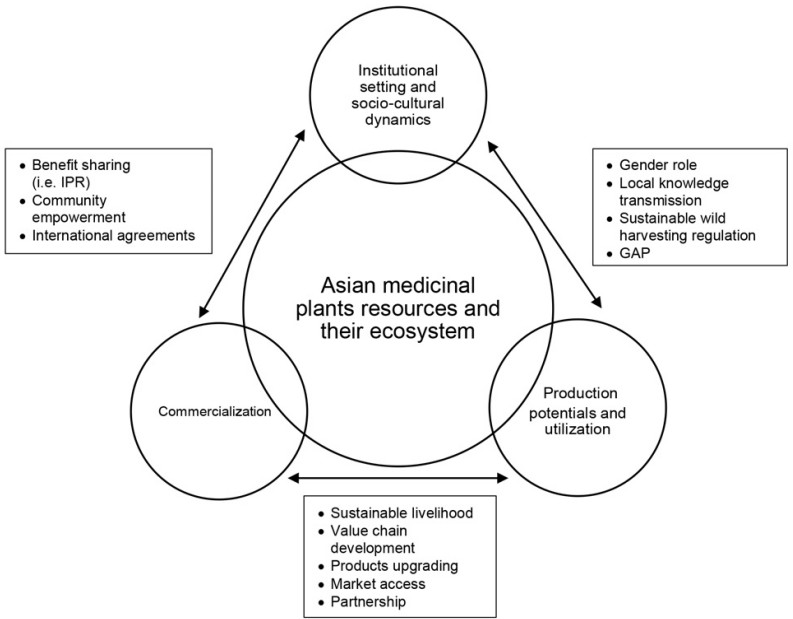

**Figure 6.** A schematic presentation of sustainable management of Asian medicinal plants.

## 5. Conclusions

Although medicinal plants research continues to receive significant attention in Asia, particularly with respect to its multiple roles in poverty alleviation and health care support, scientific information on the institutional arrangements, the potentials of different medicinal plants production systems and the utilization methods, commercialization and contribution to livelihoods remain highly fragmented. Undeniably, these issues have current and future implications for medicinal harvesting, growing, use and marketing in the context of Asia. Research gaps comprising crucial aspects of institutional success conditions, beneficial commercialization, and sustainable production systems exist in Asia. In conclusion, the following lessons are drawn from the review:

- First, there is a gradual shift from wild gathering as a production system to cultivation in Asia; nevertheless, wild collection still accounts for a significant share of medicinal plants supply. However, the economic performance of both production systems and their contribution to livelihoods still needs to be empirically clarified.
- Second, with regards to the recognition of medicinal plants potential, sub-regional variations exist. In South Asia, medicinal plants are harvested to support rural livelihood and traditional medicine systems, while in South East Asia and China, the focus is on commercialization. Information on the actual and potential contribution of medicinal plants to livelihoods remain fragmented.
- Third, in accordance with a high dependence on wild harvesting, local forest management systems such as informal rules and management practices are required to sustainably harvest medicinal plants. This will add value to MP products which will be beneficial for the economy.
- Fourth, very limited studies on the commercialization of medicinal plants exist, especially those that address the role of middlemen, boom–bust cycles, raw material readiness, and product quality.
- Fifth, medicinal plants production actors should prioritize law enforcement, benefit and knowledge sharing, and research and development.

**Author Contributions:** Conceptualization, writing—original draft preparation, methodology, funding acquisition, S.A.; supervision, writing—review and editing, J.P., J.N.K.

**Funding:** This research was funded by RISET PRO (Research and Innovation in Science and Technology Project), Ministry of Research, Technology, and Higher Education, Republic of Indonesia.

**Acknowledgments:** We would like to extend our deepest gratitude to all researchers whose articles were consulted to develop this review paper. Open Access funding was provided by the TU Dresden. We also thank all colleagues and reviewers who provided comments and feedback. Joachim Gratzfeld and Jean Linsky-BGCI are thanked for providing useful information on Asian Botanic Gardens and their medicinal plants. Lastly, we also remain thankful to Kusetiawan for assisting the figures' display, as well as Annerose Schubert and Solange and their families for the support.

**Conflicts of Interest:** The authors declare no conflict of interest.

## Appendix A

**Table A1.** Gathering and cultivation trend of medicinal plants in summary.

| Model and Findings | Selected Constraints | Suggested Way Forward | Country and References |
|---|---|---|---|
| **Wild gathering** | | | |
| Capacity building for BioTrade initiative | Over harvesting, lack of value-added activity | Trade and market development | Nepal [61] |
| Ethnomedicinal investigation in the conservation area | Unsustainable harvesting | Domestication and cultivation of selected species, household income | Bangladesh [161] |
| Examining the importance of forests to preserve medicinal plants | Illegal and over harvesting, the current policies do not meet the local people needs | Re-examine recent findings, replicating the study model, commercial cultivation | Bangladesh [11] |
| Medical material in traditional medicine | Price fluctuation, imported materials, lack of plant identification's knowledge, the limitation of wild stock, lack of research and development, strict regulation | Promoting local products, Good agricultural postharvest | Sri Lanka [268] |
| Partnership cultivation of farmers-corporation, habitat protection for sustaining wild collection and risk factor analysis | Diminished growth, increasing demand, persistent collection | Cultivation of wild types and uncommon species, market information system | Bhutan [269,270] |
| Jamu preparation as a part of Javanese health care system | Rarely cultivation, rely on small farmers and collectors supply | Ex situ and in situ conservation | Indonesia [165] |
| Traditional knowledge and conservation | Increased logging, wild harvesting, limited knowledge of young people | Knowledge transfer, the conservation of primary and river bench forest | Indonesia [164] |
| The healing art of traditional medicine | Oral knowledge, genuine knowledge is under threat | Recording traditional medicine knowledge, global collaboration, knowledge transfer | Myanmar [158] |
| Economic valuation of medicinal plants in Jagoi Community Forest | Traditional medicine is under threat, limited amount of protected species | Re-evaluating the importance of significant findings to support conservation planning | Malaysia [163] |
| Conserving Tibetan sacred mountains area | Fragile area, prone to logging, intensive | Promoting ecological and ethnobotanical uniqueness, sacred landscape | China [217] |

| | | | |
|---|---|---|---|
| (cultural and biodiversity) | interaction of human-environment | protection, putting the rural people first | |
| Co-management practice for sustainable use | Narrow marketing channel, non-grading system | Training on appropriate harvesting techniques, the dissemination of market information, customary tenure system | China [271] |
| Sustainable supply of Indian frankincense (*Boswellia serrata*) | Excessive market demand, limited supply, endangered species | Sustainable harvesting and management practices, sustainable supply chains | India [272] |
| The global trade sustainability in *Commiphora wightii* | Habitat devastation, fragmented populations, declining stocks, destructive harvesting | Immediate cultivation, endangered species consideration | India, Pakistan [273] |
| **Wild gathering/Cultivation** | | | |
| Jamu and biocultural conservation | Over utilization, traditional knowledge erosion | Ethnobotany and bioprospecting, traditional utilization, sustainable use pattern | Indonesia [52,274–277] |
| Ethnomedical plant studies in Java | Intensive practices, wild stock collection and home gardens | Research and development to produce modern medicine or drug | Indonesia [10] |
| Ecological survey for conservation and cultivation purpose in regional level, home gardens, and replanting endangered species | Sustainable harvesting awareness, appropriate training, heavily dependence on native medicinal plants | Ecological assessment integrated to cultivation, the digital database management | Vietnam [278] |
| Traditional medicinal plants survey | High demand of specific wild species, limited interest to traditional knowledge, overharvesting | Further study of local medicinal plants, cultivation, economically wild-harvested species | Vietnam [279] |
| Traditional medicinal plants utilization | Wild harvesting, herbal knowledge is under threat | Further research on bioactive compound, recording ethno-pharmacological data | Philippines [113] |
| Thai traditional medicine (TTM) application | The younger generations possess less knowledge, modern medicines influence | Preservation TTM knowledge, quantitative ethnobotany, integrating TTM to modern medicines | Thailand [280,281] |
| Production of *Swertia chirayita* from wild harvest and cultivation | Declining populations, less value addition, adulteration, | Cultivation enhancement, economic performance evaluation, trading policies | Himalaya [282] |
| Potential production systems of Paris root (*Paris polyphylla*) | Price fluctuation, limited supply capacity through cultivation, slow growing species, high market demand, unsustainable harvest | Traceable supply chains, habitat conservation, endangered species consideration | China [283,284] |
| Sustainable trade of (*Fritillaria cirrhossa*) | Limited wild stocks, extensive commercial use, low prices of cultivated bulbs, endangered species, limited supply capacity through cultivation | Trading policies, quality monitoring, habitat conservation, sustainable harvesting | China [285] |

| | | | |
|---|---|---|---|
| Assesing the Chinese medicine *Dendrobium* industry | Low level of product development, quality and efficacy, the lack of product standards | Production technology improvement, future scientific research | China [286] |
| **Cultivation** | | | |
| Exploring the management practice of selected high value species | Illegal collection and trading, no national policy to promote cultivation | Market information system, working capital mobilization for cooperative | Nepal [287] |
| Impacts assessment of income poverty and livelihood options | Training and development, enterprise establishment, marketing | Local people priorities, improved commercialization | Nepal-India [46] |
| Cultivation in harmony: local people and the Nanda Devi Biosphere Reserve | Price fluctuation, middleman and contractors are dominance | Participatory approach, women empowerment | India [49] |
| Integrated development in sustainable harvesting, cultivation, and marketing | Unregulated and overharvesting, cooperative system, minimum support for price | Product certification, extensive research on crucial part of up to down stream | India [197] |
| Cultivation for biodiversity conservation and livelihood enhancement | Lack of knowledge and overexploitation, government restrictions, limited postharvesting techniques | Scientific based for industrial needs, prioritizing marginal lands, rural technologies, intellectual property rights and benefit sharing | India [177] |
| Assessment of abundance in disturbed, undisturbed, and reforested forest, also the cultivation potential | Grazing, over gathering | Agroforestry system, restoration | Pakistan [288] |
| Cultivation experiment and economic analysis of five high market species on the mountain area | High-quality materials, education of market demand, wild harvesting | Sustainable harvesting, improving agronomic aspects | Pakistan [289] |
| Cultivation and economic evaluation of six high valuable species | Price fluctuation, production cost, variation in yield | Small-scale farming system, home garden and agroforestry | Pakistan [290] |
| Geographical distribution and conservation of a rare species (*Munronia pinnata*) | Over exploitation, sufficient planting materials | Establishing large-scale conservation, promoting intensive cultivation | Sri Lanka [291] |
| Investigating the role of medicinal plants for healthcare | Limited expertise on cultivation, unstable price, another region supply | Developing the value added, Research and development, promoting local entrepreneurs and cultivation | Lao PDR [188] |
| Proposing medicinal plants integration into forest rehabilitation | The limitation of traditional knowledge, marketing network | Proper selection of species, studying the economic potential of local herbal, propagation techniques | Malaysia [292] |

| On-farm conservation | Good Agricultural Practice attainment, edaphic factors, seed quality | Specific genetic background for long term production, environmental variations to enhance adaptation | China [293] |
|---|---|---|---|
| Community-based conservation | Unstable price, products marketing, harmony society versus materialistic-economically circumstances | Marketing cooperative, partnership, education, relevant policies, protocol guiding on the women and indigenous group, agroforestry | China [178] |
| Botanic gardens | Cultivation and propagation abilities, high-risk species identification, ex situ methods limitation | Assisting the large-scale cultivation for commercial purpose, preserving indigenous knowledge and priority species, sustainable wild harvest | South Asia and Southeast Asia [294–298] |

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
