# Peer review of "Asian Medicinal Plants’ Production and Utilization Potentials: A Review"

_sustainability, doi:10.3390/su11195483_

Round 1
Reviewer 1 Report
Comments
Ayurveda is not confined only in India, its all over the Himalaya (Nepal and India). While referring the ancient med systems, pls refer the time period when they evolved and transformed. Ayurveda(the foundation of science of life and the art of healing of Hindu culture) describes the medicinal importance of 1200 plants. The earliest record of medicinal plant use in the Himalayas is found in the Rigveda. This work was written between 4500 BC and 1600 BC, is supposed to be the oldest repository of human knowledge and describes 67 plants. https://ethnobiomed.biomedcentral.com/articles/10.1186/1746-4269-2-27 Line 55-56. This statement is very sloppy and the references used for this were old. I guess we have now updated figure like 75% or less. Please review and refer latest literature. The account of contribution of MAPs to local livelihood is worthwhile. Please elaborate the contributions of med plants at the level of subsistence or local households. Table 1. Scale of use: The scale of use could be separated by subsistence and commercial. If you can do this, lets see how the results differ. Unless, pls write this as a limitation. What about the Aichi Target. None of your literature talked about Aichi Target? Line 187. I feel skeptic in referring the term traditional medicine species, C. odorata. Please recheck this. C. odorata is native to Americas and invasive to Asia. Line 201-202. Its not always happen. It depends on species and from where to be harvested. It is also found that medicinal plants found nearby home are often collected by women whereas the timber trees often grow in distant wild forests are collected by men. Women are often associated with anthropogenic landscapes. Thus pls do not generalize the statement. Please read a paper of Bussmann et al. https://www.ncbi.nlm.nih.gov/pmc/articles/PMC5996461/This paper also talks about vertical and horizontal transmission of knowledge. Worth referring.
Line 244-245. Among the three, which one is dominant in Asia? I believe GAP is common but make sure with some references. Line 250-251. The statement is obsolete. It was happened before 20 years. Now almost of the herbs are semi-processed in the country. Line 286-287. It needs rewriting. Please make sure the Latin names are italicized. Line 393-395. While reviewing the conservation (maintaining sustainable stock) of med plants in Asia, the role of community forestry, which is highly hailed, is needed to be discussed. Pls review some stories of CF contributing the conservation of med plants in Nepal. Further discussion required on the following 1. number of cultivated species and wild collected species, 2. native medicinal plants and invasive medicinal plants and 3. The institutional initiative of one belt one road of China could be a leading factor influencing the conservation of MAPs in Asia. Intergenerational transmission. This paper talks about apprenticeship which is very important in sharing and knowledge among generation https://link.springer.com/article/10.1186/1746-4269-6-35 Line 543. Wild gathering and cultivation are complementary. They are counter-productive too. They are not complementary. Cultivation offsets wild collection pressures but the wild collections do not support cultivation. Line 601. I would rather write this sentence as ….. South East Asia and China, the focus is on commercialization of MAPs. Line 604-605. Please rewrite the sentence as …… local forest management system as informal rules alongside management practices are required to sustainably harvest the medicinal plants and value add the products if the products are used for economy.Author Response
Please see the attachment.
Reviewer 2 Report
This is an interesting and timely review focused on systematically identifying state-of-knowledge and research gaps with particular focus on institutional arrangements, production systems and potentials, and methods of utilisation and commercialization. While the manuscript has many merits, there are also some issues that need to be addressed before publication. These are specified in the following:
Major issues
Medicinal plants are not defined in the review. While there are many different definitions, the authors need to be explicit and make specific what they consider a medicinal plant (e.g. it appears they include species used as condiments, food, and cosmetics). The method description is weak. The method description – how references were located and selected – is not clear enough to allow repetition/does not provide an accurate description of what was done. Especially: (i) what were the exact search terms used (now the text just says “search terms such as …”), (ii) what are “some regional and local journals”, (iii) what were the number of references located before the application of inclusion criteria, (iv) it’s impossible to apply the inclusion criteria – they are either fuzzy (like the first criterion on sustainable development) or meaningless (the third criterion: all studies are empirical or reviews). They should instead be operationally formulated so that the reader can follow what goes on, which should allow the authors to (v) specify the number of references removed from the application of each inclusion criterion. It’s strange that the search was conducted for more than two years (line 108). What does that mean? That some terms were used at one time, and others later on? Would that explain some missing important references in the located literature, such as the Journal of Ethnopharmacology 2018 special issue on “Rising Trade, diminishing resources: dilemma on how to sustain supplies and quality of Asian Traditional Medicinals”? Lines 119-121 say “Most of the identified papers were confined to Southeast Asia, China, and South Asia. The review focus, therefore, aligns with these regions.” Given the application of the search terms, and perhaps the regional and local journals, this is not surprising. Why not just state up-front that the review is focused on these particular sub-regions (and not, for instance, central Asia or the Middle East). It’s good that Table 1 provides an overview of references allocated to main topics. The table could be improved by avoiding the same wording for Indicators and Main Topic. This indicates a lack of work in thinking through the table. It’d also be useful if the Aspect column covered the key terms mentioned in the Introduction, e.g. (i) why is “Utilization” mentioned but not “Commercialization”? (ii) the use of “Potential” carries no meaning. Lastly, the authors might wish to revisit the phrasing/naming of the Main Topics (including: (i) is “Research and development” an accurate term – why does that only apply to the Institutional aspect? (ii) is “Planting” an accurate term – why not cultivation and/or domestication? The structure and coherence of the ms can be improved by making the wordings of Aspects, Indicators, and Main Topics more accurate in Table 1, and then consequently applying the same wording as headings/sub-headings in the Result section. This would also serve to eliminate overlaps between the sub-sections in the Results, e.g. the first para in 3.1.3 does not belong to institutions but rather production systems. Once the wording in Table 1 is revised, the authors should carefully go through the Result section to make sure that all findings are in the right places. It is unclear why Table 3 is included in the main text; surely a similar table could be made for each of the main topics mentioned in Table 1. Either the inclusion of Table 3 should be justified or it should be removed (alternatively placed as an appendix, with a justification for singling out that topic). In the Discussion of wild harvesting, cultivation & domestication, and boom-bust cycles, the model proposed by Homma ((1996: Modernisation and technological dualism in the extractive economy inAmazonia. Current issues in non-timber forest products research, 59-82.) should be mentioned.
Minor issues
There are some occasional minor language glitches in the ms, e.g. (i) line 42 where as needs to be added (species used as), (ii) line 43 (and elsewhere) where "Tibetian" should be Tibetan, (iii) delete "including" in line 144, (iv) it’s two (not three) in line 196, (v) suddenly MP appears as an acronym in line 334, (vi) “und” should be “and” in Table 3, (viii) “This can of the role …” in line 462. The ms should be edited one last time. The distinction in line 44 between Tibetan medicine (China) and Sowa Rigpa (Bhutan). Sowa Rigpa is the “Tibetan science of healing” and hence the same as Tibetan medicine. The text on the inspiration for the ms (lines 84-95) is not interesting to the general reader – this is ample justification provided in the Introduction, and this section could be deleted. There are places where the language is unclear, e.g. (i) “possess their policies, and usually as the most suppliers to herbal industries” (lines 269-270), (ii) “it is occurred at around 10%” (line 290), (iii) “increase cultivation achievement at around 20% and 80% of wildcrafting” (lines 570-571). The ms should be language checked one last time. When an overview is presented, then the following sections should be in the same order, e.g. overview in lines 310-311 is not in the same order in the subsequent text. Avoid sub-conclusion on a section before the end of that section (lines 382-387 claims to be a summary of utilisation but is followed by more text on that Aspect).Author Response
Please see the attachment.

Reviewer 3 Report
General comments: the manuscript is interesting to me but needs more concrete analysis to give it a suitable paper for the ‘Sustainability’ journal. However, I’m astonished that this manuscript can be submitted in a specific medicinal plant related journal rather than in the current journal for more readers and circulation. Shorten the manuscript through sentences to draw figures and frameworks.
Comments to the authors:
First, you should clarify your priority or focused subject areas for medicinal plants research in Asian countries like only forested areas, or only homestead areas, or only indigenous people centred, or mixed of all. If your focused on mixed subjects then make a table from the existing literature to separate the literature focused. Make a table on the Asian country’s wise available literature like Southeast Asia, South Asia and China. Thus readers can easily get an idea about the research trend and track the total summary results. Results should be “Results and Discussion” to reduce the length of paper and will get more attention to the readers. Also, it will be easy to follow the paper. For institutional framework: Please draw an institutional framework of Asian medicinal plants based on the existing literature. For market analysis: Please draw a marketing structure or channel of Asian medicinal plants based on the existing literature. There is a need of trend analysis of medicinal plants research in Asian countries (e.g., Southeast Asia, South Asia and China). There is need a separate paragraph/draw a figure about the problems faced by the Asian countries (e.g., Southeast Asia, South Asia and China) to sustainable management of medicinal plants. Author’s missed a lot of interesting articles on medicinal plants (including the medicinal values of invasive alien species) in Asian countries specifically in Bangladesh like:
Rahman MH, Roy B (2014). Population Structure and Curative Uses of Invasive Plants in and around the Protected Forests of Bangladesh: A Means of Utilization of Potential Invasive Species. Journal of Ecosystems, dx.doi.org/10.1155/2014/249807. Rahman MH (2013). A Study on Exploration of Ethnobotanical Knowledge of Rural Community in Bangladesh: Basis for Biodiversity Conservation. ISRN Biodiversity, doi:10.1155/2013/369138. Khan MASA, Sultana F, Rahman MH, Roy B, Anik SI (2011). Status and ethno-medicinal usage of invasive plants in traditional health care practices: a case study from Northeastern Bangladesh. Journal of Forestry Research, 22 (4): 649-658. Rahman MH, Rahman M, Roy B, Fardusi MJ (2011). Topographical distribution, status and traditional uses of medicinal plants in a tropical forest ecosystem of North-eastern Bangladesh. International Journal of Forest Usufructs Management, 12(1): 37–56. Rana MP, Akhter F (2010). Uses of invasive alien plant species in Rema-Kalenga Wildlife Sanctuary of Bangladesh. Journal of Mountain Science 7(4): 380-385.
Conclusions should be revised and shortened. No need to repeat the method in the conclusions. Use acronyms during the first use of the words. All the scientific name of species should in Italic font. Please check. All the cited references are listed in the references section but need to check the format. Follow the attached manuscript for other comments. This article needs to be shortened and results should be present in a concrete form to follow the track of the results.

Round 2
Reviewer 3 Report
Comments to the authors:
Thanks to the authors for revised the manuscript and submitted again for another review. Most of my previous comments have been addressed except the following comments given below:
Change the sentences without specifying the name of Author. Write in third person form: The inspiration to prepare an article in this direction originates from the lead author’s experience as a plant conservation researcher in the Cibodas Botanic Garden (Indonesia) between 2008 and 2016. During this period, the lead author (Sri) was involved in ethnographic fieldwork involving medicinal plants at the buffer zone of Meru Betiri National Park (1999-2000) for a period of six months, and with the indigenous people of Tau Taa Wana (Central Sulawesi) for a four-month period (between 2010 and 2011). In Figure 2(b): specify the region wise (Southeast Asia, South Asia and China) research focus not the combined research focus. In Figure 3: specify the region wise (Southeast Asia, South Asia and China) trend of transition to medicinal plants cultivation not the combined trend. In Figure 4: should change the title of the figure. Maybe the title is: A schematic presentation of the pillars of sustainable management of Asian medicinal plants. For institutional framework: Please draw an “institutional framework of Asian medicinal plants management and conservation” based on the existing literature. For market analysis: Please draw a marketing structure or distribution channel of Asian medicinal plants based on the existing literature (i.e., production, harvest, processing, vendor, supply, end user, etc.). There is need a separate paragraph/draw a figure about the problems faced by the Asian countries (e.g., Southeast Asia, South Asia and China) to sustainable management of medicinal plants. All the cited references are listed in the references section but need to check the journal format. This article needs to be shortened (at least 20% of its long paragraphs) and results should be present in draw figures and frameworks rather than long paragraph of description. Finally, moderate English changes required. I look forward to a new revised version of the manuscript.Author Response
Dea Sir/Madam Reviewer,
We would like to send the response here and please kindly also see the attachment of the revised version of manuscript.
Thank you very much.
Sincerely yours,
On behalf of the authors
Ms. Sri Astutik
**********
Response to Reviewer 3 Comments
Comments to the authors:
Point 1:Thanks to the authors for the revised manuscript and submitting it again for another review. Most of my previous comments have been addressed except the following comments given below.
Response 1:Dear Sir/Madam. We are grateful for your valuable comments to improve this manuscript. In the whole text, we use the number of lines based on the track changes format.
Point 2:Change the sentences without specifying the name of Author.
Response 2: Thank you.Please kindly find the revised version in Lines as follows: 201, 203, 209, 280, 282, 356, 381, 405, 464-465, 537, 602, 661, 724, and 735.
Point 3: Write in third person form: The inspiration to prepare an article in this direction originates from the lead author’s experience as a plant conservation researcher in the Cibodas Botanic Garden (Indonesia) between 2008 and 2016. During this period, the lead author (Sri) was involved in ethnographic fieldwork involving medicinal plants at the buffer zone of Meru Betiri National Park (1999-2000) for a period of six months, and with the indigenous people of Tau Taa Wana (Central Sulawesi) for a four-month period (between 2010 and 2011).
Response 3: Thank you.Please kindly find the revised version in Lines 75-80.
Point 4: In Figure 2(b): specify the region wise (Southeast Asia, South Asia and China) research focus not the combined research focus.
Response 4: Thank you.We have followed your suggestion by revising Figure 2(b) in Lines 243-258.
Point 5: In Figure 3: specify the region wise (Southeast Asia, South Asia and China) trend of transition to medicinal plants cultivation not the combined trend.
Response 5: Thank you.We have followed your suggestion by revising Figure 3. It now reads Figure 5 in Lines 485-497.
Point 6: In Figure 4: you should change the title of the figure. Maybe the title is: A schematic presentation of the pillars of sustainable management of Asian medicinal plants.
Response 6: Thank you.We have revised it based on your suggestion showed in Line 803. Also, Figure 4 is now read in Figure 6.
Point 7: For institutional framework: Please draw an “institutional framework of Asian medicinal plants management and conservation” based on existing literature.
Response 7: Thank you.We appreciate it. This part would help the readers to capture the message of the institutional framework easily. Please kindly find Figure 3 in Lines 229-314.
Point 8: For market analysis: Please draw a marketing structure or distribution channel of Asian medicinal plants based on the existing literature (i.e., production, harvest, processing, vendor, supply, end user, etc.).
Response 8: Thank you.We appreciate it. The figure would help the readers to gain a better understanding of market structure rapidly. Please kindly find Figure 4 in Lines 416-441.
Point 9:There is a need for a separate paragraph/draw a figure about the problems faced by the Asian countries (e.g., Southeast Asia, South Asia and China) to sustainable management of medicinal plants.
Response 9: Thank you.We have inserted one paragraph within the Discussion section. Please kindly find it in Lines 764-774.
Point 10:All the cited references are listed in the references section but need to check the journal format.
Response 10: Thank you.Please kindly find the revised version in Pages 21-36.
Point 11:This article needs to be shortened (at least 20% of its long paragraphs) and results should be present in draw figures and frameworks rather than long paragraph of description. Finally, moderate English changes required.
Response 11: Thank you.We have tried to condense and underline the essential messages in each sub-indicator. Then, they are supported by the discussion showing the research gaps for further empirical investigations.
Point 12:I look forward to a new revised version of the manuscript.
Response 12: Thank you very much, Sir/Madam. We are delighted to resubmit the manuscript in this round. Please kindly find the attached files.
